# Hypoxia-Inducible Pathway Polymorphisms and Their Role in the Complications of Prematurity

**DOI:** 10.3390/genes14050975

**Published:** 2023-04-26

**Authors:** Ewa Strauss, Anna Gotz-Więckowska, Alicja Sobaniec, Anna Chmielarz-Czarnocińska, Dawid Szpecht, Danuta Januszkiewicz-Lewandowska

**Affiliations:** 1Institute of Human Genetics, Polish Academy of Sciences, Strzeszynska 32, 60-479 Poznan, Poland; 2Department of Ophthalmology, Poznan University of Medical Sciences, Szamarzewskiego 84, 60-569 Poznan, Poland; 3Department of Neonatology, Poznan University of Medical Sciences, Polna 33, 60-535 Poznan, Poland; 4Department of Medical Diagnostics, Poznan University of Medical Sciences, Dobra Street 38a, 60-595 Poznan, Poland; 5Department of Pediatric Oncology, Hematology and Transplantology, Poznan University of Medical Sciences, Szpitalna 27/33, 60-572 Poznan, Poland

**Keywords:** hypoxia, hypoxia at birth, *HIF1A* gene, *VEGFA* gene, prematurity

## Abstract

Excessive oxidative stress resulting from hyperoxia or hypoxia is a recognized risk factor for diseases of prematurity. However, the role of the hypoxia-related pathway in the development of these diseases has not been well studied. Therefore, this study aimed to investigate the association between four functional single nucleotide polymorphisms (SNPs) in the hypoxia-related pathway, and the development of complications of prematurity in relation to perinatal hypoxia. A total of 334 newborns born before or on the 32nd week of gestation were included in the study. The SNPs studied were *HIF1A* rs11549465 and rs11549467, *VEGFA* rs2010963, and rs833061. The findings suggest that the *HIF1A* rs11549465T allele is an independent protective factor against necrotizing enterocolitis (NEC), but may increase the risk of diffuse white matter injury (DWMI) in newborns exposed to hypoxia at birth and long-term oxygen supplementation. In addition, the rs11549467A allele was found to be an independent protective factor against respiratory distress syndrome (RDS). No significant associations with *VEGFA* SNPs were observed. These findings indicate the potential involvement of the hypoxia-inducible pathway in the pathogenesis of complications of prematurity. Studies with larger sample sizes are needed to confirm these results and explore their clinical implications.

## 1. Introduction

Preterm birth is a significant cause of perinatal mortality and morbidity, with an estimated 15 million premature births occurring annually worldwide, accounting for 11% of all births [1,2]. In Poland, 6% of all births are premature. Preterm birth is the leading cause of death among children under the age of 5, responsible for 18% of all deaths in this age group and 35% of all deaths among newborns aged <28 days [3]. Premature newborns are highly susceptible to various diseases, including necrotizing enterocolitis (NEC), bronchopulmonary dysplasia (BPD), intraventricular hemorrhage (IVH), respiratory distress syndrome (RDS), diffuse white matter injury (DWMI, previously referred to as periventricular leukomalacia), and retinopathy of prematurity (ROP) [4,5,6,7,8,9,10]. Oxidative stress, defined as an imbalance between increased reactive species and a lack of protective antioxidant capacity, has been identified as a key factor in the pathogenesis of these diseases [9].

The postnatal transition from a hypoxic intrauterine environment with 20–25 mmHg oxygen tension to an extrauterine normoxic environment of approximately 100 mmHg oxygen tension is an oxidative challenge for newborns [11,12]. Premature newborns are particularly vulnerable to oxidative damage due to deficiencies in antioxidant defenses, oxygen supplementation, immunodeficiency, and high levels of free iron [9]. Oxidative stress is especially detrimental to newborns with perinatal hypoxia-ischemia, which can result in multi-organ dysfunction and neurological deficits, leading to neonatal morbidity and mortality [13]. Pulmonary illnesses are a primary cause of postnatal hypoxia in premature infants [9,10].

Hypoxia-inducible factor-1 (HIF-1) is a crucial regulator of oxygen homeostasis, enabling adaptive responses to hypoxic environments [14]. HIF-1 functions as a heterodimer, comprising the HIF-1α and HIF-1β subunits [15]. In normoxic conditions, the activity of HIF-1 is regulated by the degradation of the HIF-1α subunit. This degradation is primarily mediated by the oxygen-dependent hydroxylation of two proline residues (Pro402 or Pro564, or both), which interact with the von Hippel–Lindau (VHL) tumor-suppressing protein. VHL recruits an E3 ubiquitin–protein ligase complex, which targets HIF-1α for proteasomal degradation [16,17]. However, under hypoxic conditions, HIF-1α is stabilized against degradation, which leads to the upregulation of numerous genes involved in various biological processes, including glycolysis, angiogenesis, and erythropoiesis [18]. Among them, the vascular endothelial growth factor gene (*VEGFA*) is a key activator of hypoxia-induced angiogenesis.

The *HIF1A* gene encodes HIF-1α and is located at chromosome 14q21-24. It consists of 15 exons and has over 450 missense variants listed in the Ensembl database. These variants are associated with various predictions of protein function, and two of them, rs11549465 (c.1744C>T, p.Pro582Ser) and rs11549467 (c.1762G>A, p.Ala588Thr), located in exon 12, have been extensively studied [19,20]. Both of these single nucleotide polymorphisms (SNPs) are situated within the oxygen-dependent degradation domain that stabilizes HIF-1α against degradation, and they exhibit higher transcriptional activity of *HIF1A* under either normoxic or hypoxic conditions [21,22]. Another synonymous variant, rs41492849 (c.1692C>T), was observed within the codon for Pro564 in the same exon, but it was extremely rare (<1%). Recently, Kuney developed a detailed profile of *HIF1A* variants and regulatory elements [23]. The *VEGFA* gene, located at chromosome 6p21.3 and consisting of eight exons exhibiting alternative splicing [24], has numerous sequence variants. However, only a few of them, located in the 5′ untranslated region or the promoter region, have been demonstrated to regulate *VEGFA* expression. For example, the rs2010963 (−634G>C) variant is associated with decreased basal promoter activity and reduced protein production, while the rs833061 (−1498C>T) variant has the opposite effect [25,26].

Recent research has identified that genetic variations in *HIF1A* are associated with over 40 different phenotypes and diseases, such as cancer, cardiovascular diseases, and metabolic disorders [20,27]. However, the potential role of *HIF1A* in the development of prematurity-related diseases has yet to be investigated. On the other hand, previous studies have suggested that *VEGFA* variants may be associated with female reproductive diseases [28,29,30], and some neonatal populations have shown potential links between *VEGFA* and the consequences of prematurity, although these have not been consistently replicated [31]. Thus, the main objective of our analysis is to comprehensively examine the associations between *HIF1A* SNPs (rs11549465 and rs11549467) and *VEGFA* SNPs (rs2010963 and rs833061) with the occurrence of NEC, BPD, IVH, RDS, DWMI, or ROP, taking into account the presence of hypoxia at birth. Our research hypothesis is that the genetically determined activity of HIF-1 and VEGF may modulate the risk of developing prematurity-related diseases in infants who have been exposed to perinatal hypoxia.

## 2. Materials and Methods

### 2.1. Study Population

This study included 334 Caucasian preterm infants who were born between 22 and less than 32 weeks of gestation (classified as very preterm and extremely preterm according to World Health Organization (WHO) criteria) [3], and who were hospitalized between 2009 and 2020 at the Gynecology and Obstetrics Clinical Hospital of Poznan University of Medical Sciences. We excluded neonates born from multiple pregnancies, those with chromosomal abnormalities or TORCH infections, and infants who did not receive antenatal steroid therapy.

We analyzed several clinical factors, including gestational age (weeks), birth weight (grams), sex, time of preterm rupture of the fetal bladder (days), mode of delivery (vaginal delivery vs. cesarean section), Apgar scores at 1 and 5 min, parameters related to oxygen therapy (such as use of surfactant, duration of mechanical ventilation (days), and total oxygen supplementation period), as well as blood transfusions. We also collected data on the presence of several diseases of prematurity, including RDS, IVH, BPD, NEC, ROP, DWMI, sepsis, and jaundice of prematurity. We diagnosed RDS, NEC, IVH, BPD, and NEC using criteria that have been described previously [32], and we diagnosed DWMI based on abnormal US or MRI results. In ROP, we distinguished a subgroup of infants with an advanced proliferative form of the disease. We determined the criteria for the diagnosis of extremely low gestational age (<28 weeks, ELGA), extremely low birth weight (<1000 g, ELBW), and intrauterine hypotrophy (fetal weight below the 10th percentile according to the percentile grids for the corresponding GA) based on WHO guidelines [3].

To determine the presence of hypoxia at birth, we assessed whether the newborn fulfilled at least two of the following criteria: acidosis (pH ≤ 7.20 in umbilical vein), Apgar score ≤6 at 5 min, and a fraction of inspired oxygen (FiO_2_) ≥0.4 needed to achieve oxygen saturation ≥86% at birth. We chose these criteria, which are less severe than those used to define neonates with birth asphyxia, according to Buonocore et al. [11] to evaluate all hypoxic infants, even those with mild hypoxia. Hypoxic infants were reanalyzed separately to verify whether stricter criteria of hypoxia, established by the American College of Obstetricians and Gynecologists (ACOG) to define newborns with birth asphyxia (pH < 7.15 in umbilical vein, Apgar score < 5 at 5 min), changed the results on the relationship between hypoxia at birth and the occurrence of diseases in preterm infants.

### 2.2. Genotyping

Genomic DNA was extracted from buccal swabs using the innuPREP DNA Kit (Analytik Jena AG, Jena, Germany) or from circulating blood lymphocytes using the QIAamp DNA Kit (Qiagen GmbH, Hilden, Germany), following the manufacturer’s instructions. Polymorphisms were determined using pre-designed TaqMan SNP genotyping assays (rs11549465: C__25473074_10, rs11549467: C__34492744_10, rs2010963: C___8311614_10, and rs833061: C___1647381_10, Thermo Fisher Scientific, Waltham, MA, USA) and the ABI 7900HT Fast Real-Time PCR System (Life Technologies, Carlsbad, CA, USA).

### 2.3. Data Analysis

Genotype frequencies were assessed for Hardy–Weinberg equilibrium using the *χ*^2^ test available at URL http://ihg.gsf.de/cgi-bin/hw/hwa1.pl (accessed on 12 December 2022). Univariate analyses were used to compare the demographic parameters of the study groups. Quantitative parameters were compared using the *t*-test or Mann–Whitney U test, and qualitative parameters using the *χ*^2^ test or Fisher’s test. The Shapiro–Wilk test was used to test the normality of the distribution. Odds ratios (ORs) and 95% confidence intervals (95%CI) were calculated for quantitative parameters. Multivariate analysis was performed using logistic regression analysis. The Bonferroni correction was used for multiple comparisons (*p*_adjusted_). Statistical analyses were carried out using Statistica v10.0 software and GraphPad Prism v6 software. Quanto software was used for power analysis. Differences were considered significant when *p* < 0.05.

## 3. Results

### 3.1. Known Risk Factors and Comorbidities

Table 1 presents detailed characteristics of the premature infants included in the study. The mean gestational age of the 334 enrolled infants was 27.6 ± 2.0 weeks, with a range between 22 and less than 32 weeks. The mean birth weight was 1114.2 ± 335.0 g, with a range of 432–2340 g. The prevalence of complications related to prematurity varied considerably, with rates ranging from 4.5% for death and 7.9% for DWMI to 23.4% for NEC, 23.4% for sepsis, 38.9% for BPD, 57.5% for IVH, 63.5% for ROP, 66.5% for RDS, and 83% for neonatal jaundice. Hypoxia at birth was found in 154 infants (46.1%), and it was associated with low gestational age and birth weight, low Apgar scores at 1 and 5 min, acidemia, high oxygen concentration in the respiratory mixture (FiO_2_ ≥ 0.4), treatment with surfactant, resuscitation, mechanical ventilation, and prolonged oxygen supply. Hypoxia at birth was also identified as a risk factor for several complications, including NEC, BPD, IVH, RDS, ROP, and DWMI, as well as death (Figure 1a).

Sixty six of the 154 hypoxic infants, which accounts for 42.9% of this subgroup, met the ACOG criteria for hypoxia. The frequencies of all studied comorbidities and death were similar in groups meeting both the broader and narrower criteria for birth hypoxia.

However, meeting the ACOG criteria was significantly associated only with BPD, IVH, RDS, ROP, and DWMI, but not NEC and death, as observed in the analyses based on the broader criteria of hypoxia (refer to Appendix A).

### 3.2. Frequency of HIF1A and VEGFA SNPs

Table 2 presents the distribution of the *HIF1A* and *VEGFA* genotypes in the entire study group. Genotyping of *HIF1A* revealed that the rs11549465 SNP occurred more frequently in the studied population than the rs11549467 variant, with observed frequencies of 0.076 for rs11549465T and 0.027 for rs11549467A. Genotyping of *VEGFA* showed a higher frequency of the rs833061 SNP than the rs2010963 variant, with observed frequencies of 0.284 for rs2010963C and 0.400 for rs833061T. The genotype distributions in the studied *HIF1A* SNPs did not deviate from the Hardy–Weinberg equilibrium. However, in the case of *VEGFA*, the distribution of the rs833061 genotypes showed an overabundance of heterozygotes (*p*_HWE_ = 0.049).

### 3.3. HIF1A and VEGFA SNPs and Comorbidities

The results of the univariate analyses are shown in Figure 2 and Appendix A. Among the *HIF1A* and *VEGFA* variants studied, only the *HIF1A* polymorphisms were found to be associated with prematurity comorbidities. Protective effects against NEC and RDS were observed for the rs11549465T and rs11549467A alleles, respectively. The rs11549465T allele was found also to be a risk factor for IVH and proliferative ROP as well as DWMI, but only in infants exposed to hypoxia at birth (OR = 3.5 (95%CI [1.3–9.0], *p* = 0.015; *p*_adjusted_ = 0.03).

The relationship between the *HIF1A* genotype, hypoxia at birth, and the development of DWMI indicated a gene–environment interaction. Coexistence of carrying the rs11549465T allele and hypoxia resulted in a 5.2-fold increase in the risk of DWMI (95%CI: 1.7–16.1; *p* = 0.008; *p*_adjusted_ = 0.016), compared to the risk of this disease in infants with the rs11549465CC genotype who were not exposed to hypoxia at birth. The observed OR value indicates an amplification of the combined effect over the additive effect of individual factors.

Further evaluation of potential factors that could modify the observed effect or more precisely predict DWMI showed that the need for prolonged mechanical ventilation could be a prognostic indicator of this disease in the studied group of infants born prematurely. The cumulative effect of the *HIF1A* rs11549465T allele, hypoxia at birth, and prolonged oxygen supplementation increased the risk of DWMI by 7.4 times (*p* = 0.005, *p*_adjusted_ = 0.015, Figure 3). ROP, BPD and IVH were the diseases most frequently comorbid with DWMI (Figure 1B).

### 3.4. Multivariate Analysis

The multivariate statistical analysis considered the *HIF1A* genotype and all potential clinical risk factors that may affect the risk of developing diseases in preterm infants. This analysis confirmed that the *HIF1A* rs11549465T allele was an independent from age, body mass and sex, and a protective factor for NEC (OR = 0.19, 95%CI [0.07–0.54], *p* = 0.002; *p*_adjusted_ = 0.01), while the rs11549467A allele was a protective factor for RDS (OR = 0.24, 95%CI [0.09–0.68], *p* = 0.007; *p*_adjusted_ = 0.035; Appendix A). The co-occurrence of the *HIF1A* rs11549465T allele with hypoxia at birth, and prolonged oxygen supply, but not gestational age, birth weight and sex, were independent risk factors for DWMI. In the regression model adjusted for other confounders, the co-occurrence of the rs11549465T allele with hypoxia at birth was associated with a 4.0-fold increase in the risk of DWMI (95%CI [1.2–12.9], *p* = 0.020). In the analysis that included only significant risk factors, the risk of this disease related to coexistence of genotype and hypoxia was increased by 4.5 times (95%CI [1.4–14.1], *p* = 0.009, *p*_adjusted_ = 0.027). No independent association was found between the *HIF1A* genotype and the development of IVH and ROP upon multivariate analysis. Low gestational age and birth weight were important risk factors for this condition.

## 4. Discussion

Hypoxia is a trigger of oxidative stress and a significant factor contributing to the pathogenesis of prematurity-related diseases. However, it remains uncertain whether this relationship is causative. In our study, we examined the associations between *HIF1A* and *VEGFA* gene variants, which act as a proxy for exposure to increased hypoxia-inducible pathway activity and angiogenesis stimulation, and the occurrence of comorbidities (NEC, BPD, IVH, RDS, DWMI, and ROP) in newborns born on or before the 32nd week of gestation. Our findings suggest that the *HIF1A* rs11549465T allele is an independent protective factor against the occurrence of NEC, but it may also increase the risk of DWMI in newborns exposed to hypoxia at birth and long-term oxygen supplementation. Additionally, the rs11549467A allele was an independent protective factor against RDS. We confirmed the role of recognized risk factors such as gestational age, birth weight, and prolonged oxygen supply in the development of prematurity-related diseases. However, we found no associations between the *VEGFA* gene variants and these diseases.

The interrelationship between gestational age, birth weight, and neonatal morbidity is well documented. The incidence of severe respiratory problems and other morbidities decreases with increasing gestational age and birth weight. The prevalence of prematurity-related diseases reported in the literature varies between European countries and the USA. For example, the incidence of NEC ranges from 2–13%, ROP from 23–52%, BPD from 10–89%, IVH from 31–36%, RDS from 7–50%, and DWMI from 8–20% [1,10]. In our study population, which consisted of infants born very preterm (58%) and extremely preterm (42%), RDS was the most frequently observed diagnosis (66.5%), followed by ROP (63.5%), IVH (57.5%), BPD (38.9%), NEC (23.4%), and DWMI (8%). These findings confirm that RDS is the most common diagnosis in infants born prematurely.

The pathogenesis of all these diseases is linked to organ immaturity, insufficient immune and antioxidant systems, environmental factors such as infections and changes in oxygen concentrations, and genetic risk factors [5,6,7,33,34,35,36,37]. Hypoxic-ischemic injury at birth affects survival and is thought to be critical for all these diseases, as observed in this study. Almost half of the newborns showed symptoms of hypoxia at birth, which were associated with death, a 3.5- to 5-fold higher risk of BPD, IVH, and ROP, or approximately 2-fold higher risk of DWMI, NEC, and RDS. The ACOG criteria for hypoxia were met by 19.8% of the studied infants, which is consistent with previous studies by Buonocore et al. [11], who found it in 20.4% of premature infants born between 26–36 weeks of gestation. Based on an analysis of the concentration of blood biomarkers of oxidative stress, including plasma levels of hypoxanthine, total hydroperoxide, and advanced oxidation protein products, the authors found that preterm newborns exposed to hypoxia are at an increased risk of oxidative stress in the first week of life. The results were consistent regardless of whether the broader or stricter (ACOG) criteria were used to identify infants at risk for hypoxia at birth. In our study, the prevalence of comorbidities was similar in the groups of hypoxic neonates selected using different criteria, indicating that the broader criteria proposed by Buonocore and colleagues were more effective in excluding cases exposed to hypoxia at birth from the control group. This result indicates a possible underestimation of the role of hypoxia at birth on the development of comorbidities associated with prematurity.

According to the literature, the strongest effect of hypoxia at birth can be expected in the case of DWMI, since the central nervous system (CNS) is highly sensitive to oxygen deficiency. During gestation, the development of the CNS is stimulated by exposure to physiological hypoxia, under which HIF-1α is stabilized and plays a crucial role in regulating neural development [38]. Premature delivery disrupts this process and can lead to the development of DWMI. Immature cerebrovascular development, birth asphyxia, and high concentrations of free ions leading to excessive free radical formation and other toxic compounds, such as peroxynitrite, are the most significant causes of injury to the (fetal) brain [13,39,40]. This study also found that prolonged oxygen supply is a subsequent predictor of this condition. It is unclear whether this relationship is causal, but previous studies also report a specific association between respiratory failure and brain injury. Therefore, continuously monitoring the respiratory rate in the neonatal intensive care unit is postulated to be useful in predicting CNS pathology [41]. A novelty with respect to previous studies is the indication of the importance of the *HIF1A* genotype in determining susceptibility to birth hypoxia-related DWMI occurrence.

The relationship between *HIF1A* alleles and the occurrence of NEC and RDS appears to involve a complex pathomechanism that encompasses various factors associated with fetal development and the activation of the inflammatory cascade by HIF-1α. Animal models of NEC suggest that altered vascular development and tone, which are components of intestinal immaturity, coupled with early exposure to cold stress and a non-milk diet, contribute to the disease [8]. *HIF1A* SNPs may affect fetal intestinal development and modulate the in utero environment by activating proinflammatory cytokines. Previous studies have linked higher concentrations of HIF-1α in amniotic fluid with a shorter amniocentesis-to-delivery interval and increased levels of IL1α, IL6, and TNFα, indicating the possibility of HIF-1α contributing to the inflammatory cascade in complicated pregnancies [42]. However, the source of HIF-1α (the mother or the fetus) remains unclear. Despite this, infection or inflammation can significantly contribute to early spontaneous preterm birth, and this adverse intrauterine environment plays a crucial role in impairing organ development, particularly the pulmonary maturation process. High levels of cytokines in the amniotic fluid lead to direct exposure of the fetal lung to proinflammatory factors, resulting in the arrest of lung development and an increased risk of respiratory diseases after birth, including RDS [43].

The opposite role of high HIF-1 activity in various diseases and conditions is a common observation. Genetically determined high activity of HIF-1, which has proangiogenic effects, benefits adaptation to hypoxia in high altitudes, ischemia related to cardiovascular disease, premature coronary artery disease, and pregnancy complicated by preeclampsia [44,45,46,47]. Under hypoxic conditions, HIF-1 activates the transcription of numerous target genes that regulate various processes, including erythropoiesis, angiogenesis, and glucose metabolism. However, exposure to elevated oxygen concentrations can cause early inactivation of these pathways in preterm infants. Additionally, exogenous oxygen administration may suppress HIF-1 activity, leading to clinically significant anemia. It is also believed that the in utero hypoxic environment protects the integrity of the intestinal epithelium, and oxygen exposure during preterm labor may increase the risk of necrotizing enterocolitis (NEC) in these infants. The HIF-1α subunit in normoxia is degraded through the ubiquitin-proteasome-prolyl hydroxylase domain (PHD) pathway; therefore, developing PHD inhibitors may be a potential strategy to pharmacologically stabilize HIF-1 to allow normal development [48,49].

On the other hand, high HIF-1 activity may be unfavorable in pathological angiogenesis, accompanying the development of solid tumors and inflammatory diseases, which may also apply to ROP. Experimental studies have demonstrated that pharmacological HIF-1 inhibition can prevent retinal neovascularization, leading to improved visual function in murine oxygen-induced ROP [50]. Anti-VEGF drugs are currently used in clinical practice to treat several types of cancers and ROP. In this study, no associations were found between *VEGFA* SNPs and the outcomes of interest. However, recent studies suggest that SNPs in this gene other than the ones examined in this analysis may have a greater impact on gene expression and protein levels [51]. Therefore, further research on this gene seems advisable.

One limitation of this study is its relatively small sample size for genetic research into DWMI. Post hoc power analysis, however, showed a statistical power of nearly 80% for the association analyses between *HIF1A* SNPs and RDS and NEC. Additionally, the results obtained met the criteria of the Bonferroni correction for multiple comparisons. The other limitation is the lack of data to assess the impact of the SNPs studied on the outcome of ROP treatment and the advancement of IVH. The latter may have contributed to the overrepresentation of IVH in the study population (57.5%) as well as the underestimation of the effect of the *HIF1A* genotype on IVH. As shown in our previous study, in a smaller group of cases in which any stage of IVH (grade I–IV in ultrasound classification) was present in 44% of infants, a stronger influence of genetic factors can be found for severe IVH (grade II–IV), occurring in approximately 60% of cases. Stage I (42% of IVH cases), defined as unilateral or bilateral reproductive matrix haemorrhage, can be excluded from association studies as a condition with low genetic impact [52]. A significant advantage in terms of the research conducted and the results obtained is that the study population is strictly defined and ethnically homogeneous.

## 5. Conclusions

In conclusion, our findings suggest a potential role of the hypoxia-inducible pathway in the pathogenesis of preterm complications, specifically DWMI, NEC, and RDS. Hypoxia at birth and prolonged oxygen supply were independent predictors of DWMI. Further studies with larger sample sizes are needed to confirm these findings and explore their clinical implications. Additionally, exploring the impact of these genetic variations on treatment outcomes may provide valuable insights for personalized management strategies for preterm infants. Studies focusing on the role of local or systemic pharmacological inhibition of HIF-1 and VEGF or, on the other hand, targeting the development of pharmacotherapy for HIF-1 stabilization and antioxidant supplementation, may be suggested based on a risk assessment that includes *HIF1A’s* genotype and clinical risk factor profile. Since even mild hypoxia has been shown to be significantly associated with complications of prematurity, more careful monitoring of oxygen supplementation could be recommended until new effective pharmacotherapy is found. Understanding the implications of our observations is of great importance for improving the care of premature babies in intensive care units.

## Figures and Tables

**Figure 1 genes-14-00975-f001:**
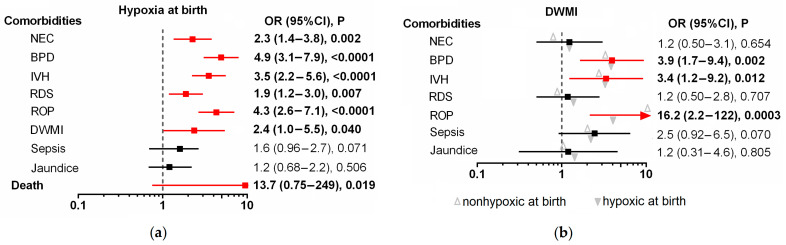
Summarized results of univariate analyses of (**a**) associations between hypoxia at birth and the occurrence of neonatal comorbidities, and (**b**) between comorbidities and presence of DWMI. Open triangles indicate non-hypoxic conditions, and solid triangles indicate hypoxic conditions.

**Figure 2 genes-14-00975-f002:**
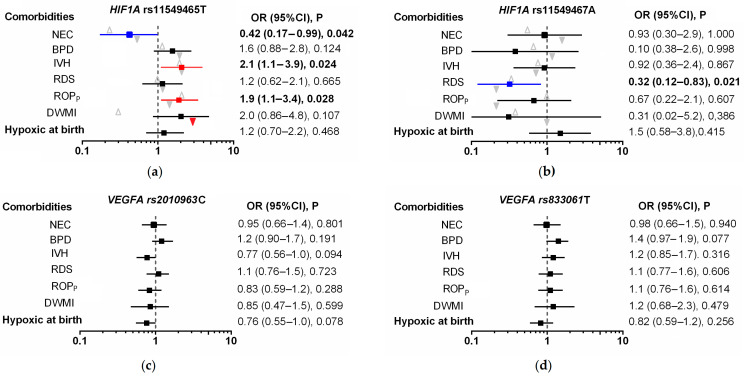
Summarized results of univariate analyses of associations between *HIF1A* (**a**,**b**) and *VEGFA* (**c**,**d**) SNPs and comorbidities and the presence of hypoxia at birth. Open triangles indicate non-hypoxic conditions, and solid triangles indicate hypoxic conditions.

**Figure 3 genes-14-00975-f003:**
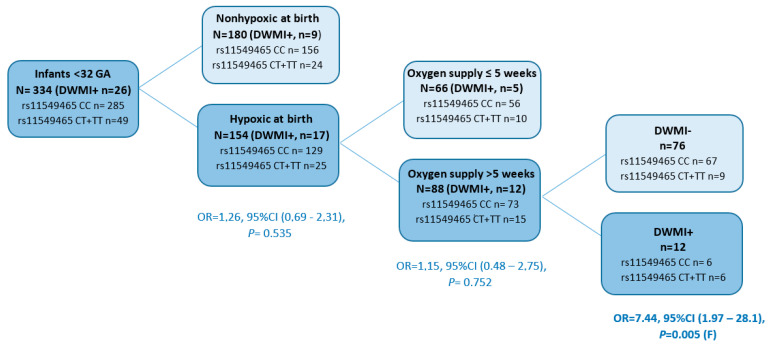
Cumulative effect of hypoxia at birth, prolonged oxygen supplementation and *HIF1A* genotype on the development of diffuse white matter injury (DWMI). The lower arm of the graph indicates that the *HIF1A rs11549465T* allele is not linked to either hypoxia at birth alone (*p* = 0.535) or in combination with oxygen supply over 5 weeks (*p* = 0.752). Only the cumulative effect of the *HIF1A* rs11549465T allele, hypoxia at birth, and prolonged oxygen supplementation increased the risk of DWMI by more than 7-fold (*p* = 0.005).

**Table 1 genes-14-00975-t001:** Demographic and clinical features of the studied group of premature infants.

Parameter		Premature Infants N = 334	Non-Hypoxic N = 180	Hypoxic N = 154	*p*
Gestational age [wk.]	M (SD)	27.6 (2.0)	28.3 (1.8)	26.9 (2.0)	<0.0001
	Range	22.0–31.9	23.0–31.9	22.0–31.9	
Body weight [g]	M (SD)	1114.2 (335.0)	1236.4 (323.2)	971.4 (289.7)	<0.0001
	Range	432–2340	570–2340	432–1900	
Intrauterine hypotrophy, n (%)		22 (6.6)	10 (5.6)	12 (7.8)	0.411
Male sex, n (%)		186 (55.7)	101 (56.1)	85 (55.2)	0.866
Risk factors at birth					
Ruptured fetal bladder, n (%)		95 (28.4)	58 (32.6)	37 (24.0)	0.085
Ruptured fetal bladder, [d], M (SD)		3.2 (9.3)	3.6 (9.4)	2.7 (9.2)	0.394
Delivery by caesarean section, n (%)		179 (53.6)	93 (51.7)	86 (55.8)	0.445
Apgar 1, Me (Q1, Q3)		5 (2, 7)	6 (5, 8)	2 (1, 5)	<0.0001
Apgar 5, Me (Q1, Q3)		7 (6, 8)	8 (7, 9)	6 (5, 7)	<0.0001
Apgar 5 ≤ 6, n (%)		120 (35.9)	8 (4.4)	112 (72.7)	<0.0001
Acidemia, n (%)		94 (28.1)	11 (6.1)	83 (53.9)	<0.0001
FiO_2_ ≥ 0.4, n (%)		227 (68.0)	80 (46.1)	147 (95.5)	<0.0001
Hypoxia at birth, n (%)		154 (46.1)	0 (0.00)	154 (100.0)	NA
Parameters related to respiratory failure					
Surfactant treatment, n (%)		156 (46.7)	59 (32.8)	97 (63.0)	<0.0001
Resuscitation, n (%)		277 (82.9)	136 (75,5)	141 (91,5)	<0.0001
Mechanical ventilation, n (%)		229 (68.6)	115 (53.0)	114 (97.44)	<0.0001
Mechanical ventilation period [d], M (SD)		22.3 (24.5)	16.8 (22.9)	28.8 (24.8)	<0.0001
Oxygen supply period [d], M (SD)		33.2 (34.7)	24.7 (30.3)	43.6 (34.8)	<0.0001
Oxygen supply >5 weeks, n (%)		141 (42.2)	53 (29.4)	88 (57.1)	<0.0001
Blood transfusions, n (%)		4.1 (3.4)	3.2 (3.6)	4.8 (3.1)	0.0005
Complications of prematurity, n (%)					
NEC		78 (23.4)	30 (16.7)	48 (31.2)	0.002
BPD		130 (38.9)	40 (22.2)	90 (58.4)	<0.0001
IVH		192 (57.5)	79 (43.9)	113 (73.4)	<0.0001
RDS		220 (66.5)	107 (59.4)	113 (73.4)	0.007
ROP		212 (63.5)	88 (47.9)	124 (80.5)	<0.0001
DWMI		26 (7.9)	9 (5.0)	17 (11.0)	0.040
Sepsis		78 (23.4)	35 (19.6)	43 (27.9)	0.068
Neonatal jaundice		278 (83.2)	147 (81.7)	130 (84.4)	0.506
Death		5 (4.5)	0 (0.0)	5 (3.2)	0.019

Abbreviations and symbols: BPD, bronchopulmonary dysplasia; DWMI, diffuse white matter injury; FiO_2_, fraction of inspired oxygen; IVH, intraventricular hemorrhage; M (SD), mean (standard deviation); Me (Q1; Q3), median (interquartile range); NEC, necrotizing enterocolitis; n (%), number (percentage); RDS, respiratory distress syndrome; ROP, retinopathy of prematurity, The presence of hypoxia at birth was recognized if newborn fulfilled at least two of the following criteria: acidemia (pH ≤ 7.20 in umbilical vein), Apgar score ≤6 at 5 min, and a fraction of inspired oxygen (FiO_2_) ≥0.4 needed to achieve oxygen saturation ≥86% at birth.

**Table 2 genes-14-00975-t002:** The distribution of the *HIF1A* and *VEGFA* genotypes in the whole group of infants.

*HIF1A* Genotype, n (%)	Premature Infants N = 334	*VEGFA* Genotype, n (%)	Premature Infants N = 334
rs11549465		rs2010963	
CC	285 (85.3)	GG	164 (49.1)
CT	47 (14.1)	GC	150 (44.9)
TT	2 (0.6)	CC	20 (5.9)
**T allele frequency**	0.076	C allele frequency	0.284
*p* _HWE_	0.967	*p* _HWE_	0.059
rs11549467		rs833061	
GG	316 (94.6)	CC	129 (38.6)
GA	18 (5.4)	CT	143 (42.8)
AA	0 (0.0)	TT	62 (18.6)
**A allele frequency**	0.027	T allele frequency	0.400
*p* _HWE_	0.613	*p* _HWE_	0.049

Abbreviations and symbols: *p*_HWE_, significance level for deviation from Hardy–Weinberg equilibrium.

## Data Availability

All data generated or analyzed during this study are included in the published article (and its Appendix A).

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
