# Peer review of "Hypoxia-Inducible Pathway Polymorphisms and Their Role in the Complications of Prematurity"

_genes, 2023, doi:10.3390/genes14050975_

Round 1

Reviewer 1 Report

A very interesting article that explores SNP polymorphisms in neonatal complications of prematurity. Few studies explore these associations and the findings are very appropiate, the interaction between environment and genetics showed, at least in part, explain a variance of comorbidities is of great interest for implementing diagnostic and follow-up policies. A great introduction, the material and methods, although concise, are well developed and the discussion follows well. Some recommendations to broaden the engagement of this article would be:

- In the title, "potential" and "pathogenesis of" could be removed.

- there are 2 tables "2". In the first table 2, write "weight" instead of "body mass".

- Figure 3 should be explained more. 

Author Response

Thank you for reading the manuscript and positive review.

Recommendation 1  In the title, "potential" and "pathogenesis of" could be removed.

Reply

According to suggestion "potential" and "pathogenesis of" were removed from the title:

Changes made -revised title:

[2-3] Hypoxia-inducible pathway polymorphisms and their role in the complications of prematurity.

Recommendation 2  there are 2 tables "2". In the first table 2, write "weight" instead of "body mass".

Reply

We have renumbered table 1 (instead of 2), and changed body mass into body weight:

Changes made

[Line 177]  Table 1. Demographic and clinical features of the studied group of premature infants

Body weight

Recommendation 3 Figure 3 should be explained more. 

According to recommendation we have added a description to Figure 3.

[229-234] Figure 3. Cumulative effect of hypoxia at birth, prolonged oxygen supplementation and HIF1A genotype on the development of diffuse white matter injury (DWMI). The lower arm of the graph indicates that the HIF1A rs11549465T allele is not linked to either hypoxia at birth alone (P=0.535) or in combination with oxygen supply over 5 weeks (P=0.752). Only the cumulative effect of the HIF1A rs11549465T allele, hypoxia at birth, and prolonged oxygen supplementation increased the risk of DWMI by more than 7-fold (P=0.005).

Reviewer 2 Report

Interesting topic, well design. The number of cases is not high but I consider that the results provided interesting conclusions.

Author Response

Thank you for reading the manuscript and positive review. We conducted additional analyzes of the results obtained in terms of corrections for multiple comparisons, which confirmed and strengthened the obtained results.

Reviewer 3 Report

The study investigated the association between four functional single nucleotide polymorphisms (SNPs) - HIF1A rs11549465 and rs11549467, VEGFA rs2010963, and rs833061 - in the hypoxia-related pathway and the development of complications of prematurity in relation to perinatal hypoxia. The study revealed that the HIF1A rs11549465T allele was an independent protective factor against necrotizing enterocolitis (NEC), but it could potentially increase the risk of diffuse white matter injury (DWMI) in newborns who had experienced hypoxia during birth and were given long-term oxygen supplementation. Similarly, the rs11549467A allele was discovered to be a distinct protective factor against respiratory distress syndrome (RDS). However, no significant associations were found between VEGFA SNPs and the studied conditions. In my view, the study is well-documented and well-structured, the provided information is valuable and up-to-date, and the conclusions are clearly articulated. I therefore recommend that the paper is accepted in its present form.  

Author Response

Thank you for reading the manuscript and positive review.

Reviewer 4 Report

Paper entitled 'Hypoxia-inducible pathway polymorphisms and their potential role in the pathogenesis of complications of prematurity' is well-designed and written. Their findings suggest that hypoxia-inducible pathway plays an important role in the pathogenesis of complications in preterm infants. I recommend authors to discuss in more details possible therapeutic strategies based on their studies in the discussion section.

Author Response

Thank you for reading the manuscript and positive review. We supplemented data about clinical relevance of our results and possible therapeutic strategies in the discussion section.

As suggested in discussion and conclusion, personalized strategies could be planned for diagnostics and treatment based on known factors such as prolonged oxygen supply, hypoxia at birth, and new factors like HIF1A genotype for the management of diseases in premature infants. The role of local or systemic pharmacological inhibition of HIF-1 or VEGF could be suggested for some diseases and the development of pharmacotherapy for the stabilization of HIF-1 for others. After revising the article, we have identified additional clinical significance of our work. We found that the broader criteria proposed by Buonocore and colleagues may be more effective than the ACOG criteria for identifying cases at risk of developing complications of prematurity due to exposure to hypoxia at birth.

Discusssion

[292-294] This result indicates a possible underestimation of the role of hypoxia at birth on the development of comorbidities associated with prematurity.

[331-341] Under hypoxic conditions, HIF-1 activates the transcription of numerous target genes that regulate various processes, including erythropoiesis, angiogenesis, and glucose metabolism. However, exposure to elevated oxygen concentrations can cause premature inactivation of these pathways in preterm infants. Additionally, exogenous oxygen administration may suppress HIF-1 activity, leading to clinically significant anemia. It is also believed that the in-utero hypoxic environment protects the integrity of the intestinal epithelium, and oxygen exposure during preterm labor may increase the risk of necrotizing enterocolitis (NEC) in these infants. The HIF-1α subunit in normoxia is degraded through the ubiquitin-proteasome-prolyl hydroxylase domain (PHD) pathway; therefore, developing PHD inhibitors may be a potential strategy to pharmacologically stabilize HIF-1 to allow normal development [48,49].

Conclusions

[374-384] Studies focusing on the role of local or systemic pharmacological inhibition of HIF-1 and VEGF or, on the other hand, targeting the development of pharmacotherapy for HIF-1 stabilization and antioxidant supplementation may be suggested based on a risk assessment including HIF1A genotype and clinical risk factor profile. Since even mild hypoxia has been shown to be significantly associated with complications of prematurity, more careful monitoring of oxygen supplementation could be recommended until new effective pharmacotherapy is found. Understanding the implications of our observations is of great importance for improving the care of premature babies in intensive care units. It is essential that resuscitation procedures consider reducing postnatal oxidative stress by using room air without adding oxygen, avoiding neonatal acidosis, and preventing sepsis.

References

  1. Park, A.M.; Sanders, T.A.; Maltepe, E. Hypoxia-inducible factor (HIF) and HIF-stabilizing agents in neonatal care. Semin Fetal Neonatal Med 2010, 15, 196-202, doi:10.1016/j.siny.2010.05.006.
  2. Hoppe, G.; Yoon, S.; Gopalan, B.; Savage, A.R.; Brown, R.; Case, K.; Vasanji, A.; Chan, E.R.; Silver, R.B.; Sears, J.E. Comparative systems pharmacology of HIF stabilization in the prevention of retinopathy of prematurity. Proc Natl Acad Sci U S A 2016, 113, E2516-2525, doi:10.1073/pnas.1523005113.

Reviewer 5 Report

This is a study on hypoxia-inducible pathway polymorphisms and their potential role in the pathogenesis of complications of prematurity.

Comments:

1.    The Ethics Committee approval is not included in the paper.

2.    Please supply the reader with a patient flowchart diagram of the population in study. How were the recruited neonates selected and how has the matching been made? Could it be a selective population?

3.    For a robust assumption the sample size should be a lot bigger. Determination of SNP’s in small sample sizes could be misleading in the final conclusion, especially if many outcomes are examined. The sample size calculation should have been made beforehand.

4.    Lines 124-129.The authors used the criteria used by Buonocore et al [11] for defining asphyxia in a small study. However the ACOG criteria are much stricter. Yet, Buonocore et al reanalyzed separately a group of asphyxiated infants (meeting stricter criteria of the ACOG) to verify whether stricter criteria of hypoxia changed the results. Maybe the authors should do the same.

5.    Line 157-158, the rates for NEC and IVH are higher than the ones presented in literature. Maybe, only IVH grade III and IV could be included.

6.    English language needs editing (Line 259, …Hypoxic-ischemic injury instead of hypoxia-ischemia injury…) (Line 119, infants instead of children.) (Line 153, please omit “a”.) (Line 162, …with resuscitation, surfactant…) (Line 230, please omit “an”.) …

7.     Line 294. To my knowledge, activation of the inflammatory cascade is mostly connected to prematurity sequelae.

8.    What is the actual clinical significance of this paper?

Author Response

  1. The Ethics Committee approval is not included in the paper.

Reply:

According to the journal's guidelines, this data is at the end of the article, not in the man text, therefore it may have gone unnoticed:

Institutional Review Board Statement: All procedures carried out on human participants in this study were in accordance with the ethical standards of the institutional and/or national research committee, and with the 1964 Helsinki declaration and its later amendments or comparable ethical standards. The study was approved by the Bioethics Committee of Poznan University of Medical Sciences (no. 1140/05 and 1117/18).

Informed Consent Statement: Informed consent was obtained from all parents of subjects involved in the study.

  1. Please supply the reader with a patient flowchart diagram of the population in study. How were the recruited neonates selected and how has the matching been made? Could it be a selective population?

Reply:

This study enrolled 334 premature infants born before 32 GA. They were consecutively recruited from 2009 to 2023 (approximately 30 cases per year) for research purposes related to obtaining scientific degrees (including PhDs for A. C.C., E.S., and D.S., and a professorship for A.G.W.). Infants were included based on matching inclusion and exclusion criteria (born before 32 weeks gestation and no multiple pregnancies, chromosomal abnormalities, or TORCH infections, and received antenatal steroid therapy) and with parental consent for genetic testing and storage of samples for future research, as described in the methods section, as well as Institutional Review Board and Informed Consent Statements.

Hypoxia at birth was not included in the matching criteria (cross-sectional study of the cohort of infants), therefore we believe that the other variables are distributed similarly to the population of prematurely born children. As a statistical sample of the population was not feasible (this study is not a randomized clinical trial), odds ratios were used as a measure of the strength of association.

The flowchart diagram in this case, it does not bring any new information beyond inclusion and exclusion criteria,  described in methods. The flowchart diagram we presented in details in our publication for IJMS, which addresses the incidence, treatment and treatment outcomes of retinopathy of prematurity and explains cases lost to follow-up. (ijms-2308718 SELENOP rs3877899 variant affects the risk of developing advanced stages of retinopathy of prematurity (ROP), which is pending Editor decision after minor revisions).

  1. For a robust assumption the sample size should be a lot bigger. Determination of SNP’s in small sample sizes could be misleading in the final conclusion, especially if many outcomes are examined. The sample size calculation should have been made beforehand.

Reply:

We agree that the sample size could be larger, especially if many outcomes are examined, as we described this in the limitations of the study. Preliminary analyses of statistical power were also carried out for the diseases studied, such as ROP and RDS, but the result obtained for DWMI was a new one that we did not expect. Post-hoc analyses were performed and presented in accordance with current methodology. There were also many additional analyses performed before preparing the manuscript to make sure that our results are not random. 

For example we examined our results for multiple comparisons, and found that they met even the most restrictive adjustment (the Bonferroni correction for multiple comparisons). We add this information in the manuscript, for following analyses a) the association between rs11549465 SNP and the occurrence of DWMI in cases exposed to hypoxia at birth; b) the cumulative effect of: hypoxia at birth, prolonged oxygen supplementation and rs11549465 SNP on DWMI c) the multivariate analyses.

The multivariate analysis was also supplemented with Model 2 that includes only independent risk factors for studied diseases (Supplementary Table S4). In the first attempt we decided to show the results for all factors that can be important for the occurrence of diseases in premature infants, in order to the reader can compare their effects between studied diseases on their own (Model 1). We think that Model 2 is also important.

Moreover, it should be noted that this well-characterized population of preterm infants is a relatively large group for studies of associations. The study population is strictly defined and ethnically homogeneous, which is a significant advantage in terms of the research conducted and the results obtained. For comparison, previous genome-wide association study (GWAS)  on Chorioamnionitis among Preterm Infants [PMID: 30674050; Am J Perinatol 2019], include 213 clinical chorioamnionitis cases and 707 clinically uninfected controls, however only 58 (27.2%) cases and 165 (23.3%) controls from Caucasian population (N=223). The whole group, was 2,75-times larger from ours in simultaneous analysis of thousands of SNPs.

Changes:

Methods

[149-150] The Bonferroni correction was used for multiple comparisons (Padjusted).

3.3. HIF1A and VEGFA SNPs and comorbidities

[207-209] The rs11549465T allele was found to be also a risk factor for IVH, and proliferative ROP, as well as DWMI, but only in infants exposed to hypoxia at birth (OR=3.5 (95%CI[1.3-9.0], P=0.015; Padjusted=0.03).

[212-214] Coexistence of carrying the rs11549465T allele and hypoxia resulted in a 5.2-fold increase in the risk of DWMI (95%CI:1.7-16.1; P=0.008; Padjusted =0.016), compared to the risk of this disease in infants with the rs11549465CC genotype who were not exposed to hypoxia at birth

 3.3. Multivariate analysis

[235-248] The multivariate statistical analysis considered the HIF1A genotype and all potential clinical risk factors that may affect the risk of developing diseases in preterm infants. This analysis confirmed that the HIF1A rs11549465T allele was an independent from age, body mass and sex, protective factor for NEC (OR=0.19, 95%CI [0.07-0.54], P=0.002; Padjusted=0.01), while the rs11549467A allele was a protective factor for RDS (OR=0.24, 95%CI [0.09-0.68], P=0.007; Padjusted=0.035; Supplemental Table S4). The co-occurrence of the HIF1A rs11549465T allele with hypoxia at birth, and prolonged oxygen supply, but not gestational age, birth weight and sex were independent risk factors for DWMI. In the regression model adjusted for other confounders, the co-occurrence of the rs11549465T allele with hypoxia at birth was associated with a 4.0-fold increase in the risk of DWMI (95%CI [1.2-12.9], P=0.020). In the analysis that included only significant risk factors, the risk of this disease related to coexistence of genotype and hypoxia was increased by 4.5 times (95%CI [1.4-14.1], P=0.009, Padjusted=0.027).

 Discussion (limitation section):

[354-355] Additionally the results obtained met the criteria of the Bonferroni correction for multiple comparisons.

[364-365] A significant advantage in terms of the research conducted and the results obtained is that the study population is strictly defined and ethnically homogeneous.

Supplemental material

We have added Model 2 (statistically significant risk factors ) in Table S4 and the information which factors : *- met the criteria of the Bonferroni correction for multiple comparisons

  1. Lines 124-129.The authors used the criteria used by Buonocore et al [11] for defining asphyxia in a small study. However the ACOG criteria are much stricter. Yet, Buonocore et al reanalyzed separately a group of asphyxiated infants (meeting stricter criteria of the ACOG) to verify whether stricter criteria of hypoxia changed the results. Maybe the authors should do the same.

Reply:            

Buonocore et al (2002) analyzed 3 blood biomarkers of oxidative stress, namely plasma levels of hypoxanthine, total hydroperoxide (TH), and advanced oxidation protein products (AOPP) in cord blood and blood drawn on 7th day of life. The study group included 49 newborns of gestational age 26–36 wk, among them 34 (69,4%) were hypoxic at birth according to wider criteria, and 10 ( 20.4% of the group) met stricter criteria of hypoxia (pH below/equal 7.15 in umbilical vein, Apgar score below/equal 5 at 5 min;). The results according oxidative stress in this subgroup (in 10 of 34 hypoxic babies meeting stricter criteria of hypoxia with respect to nonhypoxic babies) were similar.

In our group (22-32 wk),  very similar percentage of cases with asphyxia was present. 19,8% ( 66 cases were recognized to met ACOG criteria). We verify data for associations between hypoxia at birth (stricter criteria of the ACOG) and the occurrence of neonatal comorbidities (Supplementary Table S1). The importance of asphyxia for the occurrence of BPD, IVH, RDS, ROPp, DWMI has been confirmed. The criteria developed by Buonocore et al were found to better indicate the risk of complications of prematurity related to hypoxia at birth.

Changes made

[2.1. Study population]

[128-132] Hypoxic infants were reanalyzed separately to verify whether stricter criteria of hypoxia, established by the American College of Obstetricians and Gynecologists (ACOG) to define newborns with birth asphyxia (pH<7.15 in umbilical vein, Apgar score<5 at 5 min) changed our results on the relationship between hypoxia and the occurrence of diseases in preterm infants.

[3.3. HIF1A and VEGFA SNPs and comorbidities]

[173-175][188-190] Sixty-six out of 154 hypoxic infants, which accounts for 42.9% of this subgroup, met the ACOG criteria for hypoxia. The frequencies of all studied comorbidities and death, were similar in groups meeting broader and narrower criteria for birth hypoxia. However, meeting the ACOG criteria was significantly associated only with BPD, IVH, RDS, ROP, and DWMI, but not NEC and death, as observed in the analyses based on the broader criteria of hypoxia (refer to Supplementary Table S1).

[Discussion]

[281-294]

The ACOG criteria for hypoxia were met by 19.8% of the studied infants, which is consistent with previous studies by Buonocore et al. [11], who found it in 20.4% of premature infants born between 26-36 weeks of gestation. Based on an analysis of the concentration of blood biomarkers of oxidative stress, including plasma levels of hypoxanthine, total hydroperoxide, and advanced oxidation protein products, the authors found that preterm newborns exposed to hypoxia are at an increased risk of oxidative stress in the first week of life. The results were consistent regardless of whether the broader or stricter (ACOG) criteria were used to identify infants at risk for hypoxia at birth. In our study, the prevalence of comorbidities was similar in the groups of hypoxic neonates selected using different criteria, indicating that the broader criteria proposed by Buonocore and colleagues were more effective in excluding cases exposed to hypoxia at birth from the control group. This result indicates a possible underestimation of the role of hypoxia at birth on the development of comorbidities associated with prematurity.

[Supplementary data]:

Supplementary Table S1: Reanalysis of a group of infants who meet stricter ACOG criteria for hypoxia.

[other tables have been renumbered]

  1. Line 157-158, the rates for NEC and IVH are higher than the ones presented in literature. Maybe, only IVH grade III and IV could be included.

Reply

Unfortunately, not all newborns have complete data on the severity of the disease. This may have contributed to the overrepresentation of IVH in the study population and the underestimation of the effect of HIF1A genotype on IVH. We previously observed, in a group of cases characterized in terms of staging, stronger influence of genetic factors in severe IVH, grade II-IV. [Kosik, K et all, Childs Nerv Syst 2023]. We have added this information in the study limitations section. When collecting new cases, we take into account the extended IVH clinical description.

Corrections in text:

[356-365] The other limitation is the lack of data to assess the impact of the SNPs studied on the outcome of ROP treatment and the advancement of IVH. The latter may have contributed to the overrepresentation of IVH in the study population (57.5%) as well as the underestimation of the effect of HIF1A genotype on IVH. As shown in our previous study, in a smaller group of cases where any stage of IVH (grade I-IV in ultrasound classification) was present in 44% of infants, stronger influence of genetic factors can be found for severe IVH (grade II-IV), occurred in approximately 60% of cases. Stage I (42% of IVH cases), defined as unilateral or bilateral reproductive matrix haemorrhage, can be excluded from association studies as a condition with low genetic impact. [52].

References

  1. Kosik, K.; Szpecht, D.; Karbowski, Ł.; Al-Saad, S.R.; Chmielarz-Czarnocińska, A.; Minta, M.; Sowińska, A.; Strauss, E. Hemangioma-related gene polymorphisms in the pathogenesis of intraventricular hemorrhage in preterm infants. Childs Nerv Syst 2023, doi:10.1007/s00381-023-05824-4.

  1. English language needs editing(Line 259, …Hypoxic-ischemic injury instead of hypoxia-ischemia injury…) (Line 119, infants instead of children.) (Line 153, please omit “a”.) (Line 162, …with resuscitation, surfactant…) (Line 230, please omit “an”.) …

We corrected English (original numbering):

(Line 259) Hypoxic-ischemic injury instead of hypoxia-ischemia injury

(Line 119 and the whole text without Introduction “children under the age of 5” ) infants instead of children

(Line 153) , we omitted “a”.

(Line 162, the order of words has been changed: …with resuscitation, surfactant…)

(Line 230, we omitted “an”.) …

  1. Line 294. To my knowledge, activation of the inflammatory cascade is mostly connected to prematurity sequelae.

Early preterm births are often associated with intrauterine infections, which trigger an inflammatory response believed to cause preterm labor and result in fetal growth retardation and injury to the developing fetal lung and brain (cited in 23; Adams Waldorf et al Reproduction, 2013). The sensitivity to these agents may depend on the fetal HIF1A genotype.

In this study, we excluded pregnancies with the classic group of teratogenic pathogens known as “TORCH” (Toxoplasma gondii, Treponema pallidum, Rubella virus, Cytomegalovirus, and Herpes simplex virus), which could be a confounding factor for these results and have a stronger effect than the infant's genotype. The impact of lower genital tract bacteria, other viruses, and temporary infections can be assumed. The mechanisms of action of pathogens are best described for teratogenic pathogens, and similar effects can be postulated based on this information.

  1. Adams Waldorf, K.M. and R.M. McAdams, Influence of infection during pregnancy on fetal development. Reproduction, 2013. 146(5): p. R151-62.

  1. What is the actual clinicalsignificance of this paper?

As suggested in discussion and conclusion, personalized strategies could be planned for diagnostics and treatment based on known factors such as prolonged oxygen supply, hypoxia at birth, and new factors like HIF1A genotype for the management of diseases in premature infants. The role of local or systemic pharmacological inhibition of HIF-1 or VEGF could be suggested for some diseases and the development of pharmacotherapy for the stabilization of HIF-1 for others. After revising the article, we have identified additional clinical significance of our work. For instance, we found that the broader criteria proposed by Buonocore and colleagues may be more effective than the ACOG criteria for identifying cases at risk of developing complications of prematurity due to exposure to hypoxia at birth.

Corrections in text:

Discusssion

[292-294] This result indicates a possible underestimation of the role of hypoxia at birth on the development of comorbidities associated with prematurity.

[331-341] Under hypoxic conditions, HIF-1 activates the transcription of numerous target genes that regulate various processes, including erythropoiesis, angiogenesis, and glucose metabolism. However, exposure to elevated oxygen concentrations can cause premature inactivation of these pathways in preterm infants. Additionally, exogenous oxygen administration may suppress HIF-1 activity, leading to clinically significant anemia. It is also believed that the in-utero hypoxic environment protects the integrity of the intestinal epithelium, and oxygen exposure during preterm labor may increase the risk of necrotizing enterocolitis (NEC) in these infants. The HIF-1α subunit in normoxia is degraded through the ubiquitin-proteasome-prolyl hydroxylase domain (PHD) pathway; therefore, developing PHD inhibitors may be a potential strategy to pharmacologically stabilize HIF-1 to allow nor-mal development [48,49].

Conclusions

[374-384] Studies focusing on the role of local or systemic pharmacological inhibition of HIF-1 and VEGF or, on the other hand, targeting the development of pharmacotherapy for HIF-1 stabilization and antioxidant supplementation may be suggested based on a risk assessment including HIF1A genotype and clinical risk factor profile. Since even mild hypoxia has been shown to be significantly associated with complications of prematurity, more careful monitoring of oxygen supplementation could be recommended until new effective pharmacotherapy is found. Understanding the implications of our observations is of great importance for improving the care of premature babies in intensive care units. It is essential that resuscitation procedures consider reducing postnatal oxidative stress by using room air without adding oxygen, avoiding neonatal acidosis, and preventing sepsis.

References

  1. Park, A.M.; Sanders, T.A.; Maltepe, E. Hypoxia-inducible factor (HIF) and HIF-stabilizing agents in neonatal care. Semin Fetal Neonatal Med 2010, 15, 196-202, doi:10.1016/j.siny.2010.05.006.
  2. Hoppe, G.; Yoon, S.; Gopalan, B.; Savage, A.R.; Brown, R.; Case, K.; Vasanji, A.; Chan, E.R.; Silver, R.B.; Sears, J.E. Comparative systems pharmacology of HIF stabilization in the prevention of retinopathy of prematurity. Proc Natl Acad Sci U S A 2016, 113, E2516-2525, doi:10.1073/pnas.1523005113.

Round 2

Reviewer 5 Report

1.       Line 100: 'infants’ instead of children

2.       Line 345: ‘early’ instead of premature

3.       Lines 394-396: Resuscitation using room-air is used only in term infants. In preterm infants it is recommended starting by a specific oxygen-level supplementation (i.e. 30%) and adjust according to oxyhemoglobulin saturation(SatO2). Therefore, these lines should be excluded.

Author Response

Thank you for reading the manuscript and providing the recommendations.

Changes made to the manuscript genes-2284299

  1. Line 100:'infants’ instead of children

We corrected according recommendation

  1. Line 345:‘early’ instead of premature

Line 334- We changed premature into early

However, exposure to elevated oxygen concentrations can cause (premature) early inactivation of these pathways in preterm infants.

  1. Lines 394-396:Resuscitation using room-air is used only in term infants. In preterm infants it is recommended starting by a specific oxygen-level supplementation (i.e. 30%) and adjust according to oxyhemoglobulin saturation(SatO2). Therefore, these lines should be excluded.

We excluded:

It is essential that resuscitation procedures consider reducing postnatal oxidative stress by using room air without adding oxygen, avoiding neonatal acidosis, and preventing sepsis.